# Chilensosides E, F, and G—New Tetrasulfated Triterpene Glycosides from the Sea Cucumber *Paracaudina chilensis* (Caudinidae, Molpadida): Structures, Activity, and Biogenesis

**DOI:** 10.3390/md21020114

**Published:** 2023-02-05

**Authors:** Alexandra S. Silchenko, Sergey A. Avilov, Roman S. Popov, Pavel S. Dmitrenok, Ekaterina A. Chingizova, Boris B. Grebnev, Anton B. Rasin, Vladimir I. Kalinin

**Affiliations:** G.B. Elyakov Pacific Institute of Bioorganic Chemistry, Far Eastern Branch of the Russian Academy of Sciences, Pr. 100-letya Vladivostoka 159, 690022 Vladivostok, Russia

**Keywords:** *Paracaudina chilensis*, Caudinidae, Molpadida, triterpene glycosides, chilensosides, sea cucumber, hemolytic and cytotoxic activity

## Abstract

Three new tetrasulfated triterpene glycosides, chilensosides E (**1**), F (**2**), and G (**3**), have been isolated from the Far-Eastern sea cucumber *Paracaudina chilensis* (Caudinidae, Molpadida). The structures were established based on extensive analysis of 1D and 2D NMR spectra and confirmed by HR-ESI-MS data. The compounds differ in their carbohydrate chains, namely in the number of monosaccharide residues (five or six) and in the positions of sulfate groups. Chilensosides E (**1**) and F (**2**) are tetrasulfated pentaosides with the position of one of the sulfate groups at C-3 Glc3, and chilensoside G (**3**) is a tetrasulfated hexaoside. The biogenetic analysis of the glycosides of *P. chilensis* has revealed that the structures form a network due to the attachment of sulfate groups to almost all possible positions. The upper semi-chain is sulfated earlier in the biosynthetic process than the lower one. Noticeably, the presence of a sulfate group at C-3 Glc3—a terminal monosaccharide residue in the bottom semi-chain of compounds **1** and **2**—excludes the possibility of this sugar chain’s further elongation. Presumably, the processes of glycosylation and sulfation are concurrent biosynthetic stages. They can be shifted in time in relation to each other, which is a characteristic feature of the mosaic type of biosynthesis. The hemolytic action of compounds **1**–**3** against human erythrocytes and cytotoxic activities against five human cancer cell lines were tested. The compounds showed moderate hemolytic activity but were inactive against cancer cells, probably because of their structural peculiarities, such as the combination of positions of four sulfate groups.

## 1. Introduction

Triterpene glycosides from sea cucumbers are long-studied metabolites that still draw interest from researchers from different scientific fields. The structural studies of the compounds from holothuroids—representatives of different taxonomic groups—are predominating [1,2,3,4], but they are in close connection with biological activity research [4,5,6,7,8] and, therefore, structure-activity relationships analyses [3,9]. An additional direction of the glycosides’ applied investigations is their use as chemotaxonomic markers to clarify the systematic position of the producing species of sea cucumbers [10,11,12]. This issue is also relevant for the species under investigation—*Paracaudina chilensis*, belonging to the family Molpadida. The systematics and phylogeny of this taxonomic group have raised questions until now [13,14]. Our recent study concerning the isolation and structural elucidation of chilensosides A–D—the glycosides from *P. chilensis*—demonstrated their structural similarity to the glycosides isolated from different representatives of the order Dendrochirotida, indicating the closeness of Molpadida to Dendrochirotida [15]. Chilensosides A–D contains two different aglycones and four types of carbohydrate chains, differing in the positions and quantity of sulfate groups from two (chilensosides A, A_1_, B) to three in chilensoside C and four in chilensoside D. Three out of five glycosides demonstrated relatively high hemolytic and cytotoxic activity.

In continuation of the structural research on the glycosides from *P. chilensis,* the isolation, structure elucidation, biologic activity testing, and carbohydrate chain biogenesis of new sulfated highly polar glycosides chilensosides E (**1**), F (**2**), and G (**3**) are reported. Each of the three new compounds contains four sulfate groups. The first finding of two tetrasulfated glycosides in sea cucumbers has occurred fairly recently in *Psolus fabricii* [16]. Three years later, three other glycosides containing four sulfate groups were found in the sea cucumber *Psolus chitonoides* [17] and one in *P. chilensis* [15]. The finding of tetrasulfated glycosides in recent years could be explained by the increased ability of HPLC separation techniques, including the use of different sorbents (immobile phases) to separate earlier inseparable polar substances [18]. The expansion of knowledge concerning the structural variability of the glycosides enables the clarification of the biosynthetic pathways of these metabolites.

The chemical structures of **1**–**3** were elucidated by the analyses of the ^1^H, ^13^C NMR, 1D TOCSY, and 2D NMR (^1^H,^1^H-COSY, HMBC, HSQC, ROESY) spectra, as well as HR-ESI mass spectra. All the original spectra are displayed in Appendix A. The hemolytic activity on human erythrocytes and cytotoxic activities on leukemia promyeloblast HL-60, adenocarcinoma HeLa, colorectal adenocarcinoma DLD-1, human neuroblastoma SH-SY5Y, and monocytic THP-1 cells were tested.

## 2. Results and Discussion

### 2.1. Structure Elucidation of the Glycosides

The crude glycosidic mixture of *Paracaudina chilensis* was isolated by hydrophobic chromatography of the concentrated ethanolic extract on a Polychrom-1 column (powdered Teflon, Biolar, Latvia). Its further separation using chromatography on Si gel columns with a stepped gradient of the system of eluents CHCl3/EtOH/H2O in the ratios of 100:100:17, 100:125:25, and 100:150:50 was followed by the additional purification of the obtained fractions, yielding the subfractions I.0, I.1, II, III.1, and III.2. The individual glycosides **1**–**3** (Figure 1) have been isolated using HPLC of subfractions II and III.2 on silica-based columns Supelcosil LC-Si (4.6 *×* 150 mm), and reversed-phase semipreparative columns Supelco Ascentis RP-Amide (10 *×* 250 mm) and Diasfer 110 C-8 (4.6 *×* 250 mm).

The sugar configurations in the glycosides **1**–**3** were assigned as *D*, along with the biogenetic analogies with all other known triterpene glycosides from the sea cucumber.

The holostane aglycones of chilensosides E (**1**), F (**2**), and G (**3**) (Table 1, Appendix A) have 9(11)- and 24(25)-double bonds as well as a 16-oxo-group, and are identical to each other and those of chilensosides A_1_, B, C, and D, isolated earlier [15]. The identity of the aglycones of compounds **1**–**3** was evidenced by the coincidence of their NMR spectroscopic data. The structure of the aglycone moiety of **1** was established based on 2D NMR spectra analyses, where the characteristic features were found: 18(20)-lactone signals at δ_C_ 176.9 (C-18) and δ_C_ 83.1 (C-20), 9(11)-double bond signals at δ_C_ 151.1 (C-9), δ_C_ 111.3 (C-11) and δ_H_ 5.37 (brs, H-11), and the downfield signal of quaternary carbon at δ_C_ 214.6 that corresponded to a carbonyl group at C-16. This position was corroborated by the cross-peaks H_2_-15/C-16 and H-17/C-16 in the HMBC spectrum of **1**. The signals characteristic of 24(25)-double bonds was observed in the downfield region of the ^13^C NMR spectrum at δ_C_ 124.1 (C-24) and 132.1 (C-25). The correlations between H-24/H-22, H-26/H-24, and H-27/H-23 in the ROESY spectrum and H-26/C: 24, 25, and 27 cross-peaks in the HMBC spectrum confirmed the structure of the side chain (Figure 2).

Extensive analysis of the ^1^H,^1^H-COSY, 1D TOCSY, HSQC, and ROESY spectra of carbohydrate parts of compounds **1**–**2** (Table 2 and Table 3) indicated the same monosaccharide composition: one xylose (Xyl1), one quinovose (Qui2), two glucose (Glc3 and Glc4), and 3-*O*-methylglucose (MeGlc5) residues. The positions of glycosidic linkages established by the ROESY and HMBC (Figure 2) correlations were typical for the sea cucumber glycosides (*β*-(1→4) and *β*-(1→3) bonds) and corresponded to the oligosaccharide chains branched by C-4 Xyl1 having a bottom semi-chain composed of Qui2 and Glc3 and an upper semi-chain composed of Xyl1, Glc4, and MeGlc5.

The molecular formula of chilensoside E (**1**) was determined to be C_60_H_90_O_39_S_4_Na_4_ from the [M_4Na_–2Na]^2−^ ion peak at *m*/*z* 804.1857 (calc. 804.1874), [M_4Na_–3Na]^3−^ ion peak at *m*/*z* 528.4622 (calc. 528.4619), and [M_4Na_–4Na]^4−^ ion peak at *m*/*z* 390.5999 (calc. 390.5991) in the (−)HR-ESI-MS (Appendix A). The ^1^H and ^13^C NMR spectra of the carbohydrate chain of chilensoside E (**1**) (Table 2, Appendix A) demonstrated five characteristic doublets of anomeric protons at δ_H_ 4.61–5.16 (*J* = 7.4–8.3 Hz) and the signals of anomeric carbons at δ_C_ 103.4–104.6, indicating the presence of a pentasaccharide chain and *β*-configurations of glycosidic bonds. The MS data indicated the presence of four sulfate groups in the sugar chain of **1** due to the registration of a four-charged ion. The analysis of the signals of the isolated spin system corresponding to the Glc3 residue deduced by 1D TOCSY and ^1^H,^1^H-COSY spectra showed three strongly deshielded proton signals (δ_H_ 4.98 (H-3 Glc3); 4.91 and 4.52 (2H-6 Glc3)). The HSQC spectrum correlated them with two carbon signals at δ_C_ 83.8 (C-3 Glc3) and 67.4 (C-6 Glc3). These data indicated that two sulfate groups were linked to the Glc3 residue at C-3 and C-6. The ROESY and HMBC correlations of H-3 Glc3 with any protons or carbons of neighboring monosaccharide residues were absent. These data indicated that Glc3 is a terminal residue of the bottom semi-chain. The attachment of the third sulfate group to C-6 Glc4 was deduced from the characteristic signal at δ_C_ 67.7 assigned in the same manner as for Glc3. The position of the last sulfate group was established to be at C-4 MeGlc5 due to the deshielding of its signal to δ_C_ 76.1 when compared with corresponding signals in the spectra of chilensosides A and C observed at δ_C_ 70.0 [15]. Moreover, all the signals of this monosaccharide unit in the ^13^C NMR spectrum of **1** coincided with the signals of 3-*O*-methylglucose residue, sulfated by C-4, in the spectrum of chilensoside B [15]. Hence, chilensoside E (**1**) is characterized by a tetrasulfated carbohydrate chain with a new combination of sulfate group positions.

The (*−*)ESI-MS/MS of **1** (Appendix A) demonstrated the fragmentation of [M_4Na_–2Na]^2−^ ion with *m*/*z* 804.2 giving the fragment ion-peak at *m*/*z* 665.7 [M_4Na_–2Na–MeGlcSO_3_Na+H]^2−^, as well as the fragmentation of [M_4Na_–3Na]^3−^ ion with *m*/*z* 528.5 resulting in the ion peaks [M_4Na_–3Na–HSO_4_Na–SO_3_Na+2H]^3−^ and [M_4Na_–3Na–MeGlcSO_3_Na+H]^3−^ observed at *m*/*z* 456.8 and 430.1, correspondingly.

These data indicate that chilensoside E (**1**) is 3*β*-*O*-{3,6-*O*-sodium disulfate-*β*-D-glucopyranosyl-(1→4)-*β*-D-quinovopyranosyl-(1→2)-[4-*O*-sodium sulfate-3-*O*-methyl-*β*-D-glucopyranosyl-(1→3)-6-*O*-sodium sulfate-*β*-D-glucopyranosyl-(1→4)]-*β*-D-xylopyranosyl}-16-oxoholosta-9(11),24(25)-diene.

The molecular formula of chilensoside F (**2**) was determined to be C_60_H_90_O_39_S_4_Na_4_ from the [M_4Na_–2Na]^2−^ ion peak at *m*/*z* 804.1875 (calc. 804.1874), [M_4Na_–3Na]^3−^ ion peak at *m*/*z* 528.4628 (calc. 528.4619) and [M_4Na_–4Na]^4−^ ion peak at *m*/*z* 390.5998 (calc. 390.5991) in the (−)HR-ESI-MS (Appendix A), indicating compound **2** is isomeric to **1** presumably by the sulfate group positions. The carbohydrate chain of **2** consisted of five sugar residues (from five signals of anomeric protons at δ_H_ 4.62–5.20 (*J* = 7.3–8.2 Hz)) (Table 3, Appendix A). Two glucose residues (Glc3 and Glc4) were sulfated by C-6, which was deduced from the presence of characteristic signals of the sulfated hydroxy methylene groups at δ_C_ 67.4 and 68.5, while the 3-*O*-methylglucose residue did not bear any sulfates (δс 70.0 (C-4 MeGlc5) and δс 62.0 (C-6 MeGlc5)). An additional two sulfates were attached to C-3 Glc3 and C-4 Glc4, deduced from the downfield shifting of their signals to 83.9 and 75.5, respectively. The comparison of the signals corresponding to the Glc3 residue in the ^13^C NMR spectra of chilensosides E (**1**) and F (**2**) showed their coincidence corroborating the presence of sulfate groups at C-3 Glc3 and C-6 Glc3 in **2**. The same procedure was conducted for the signals of the Glc4 residue in the ^13^C NMR spectra of chilensosides A, A_1_, C [15], and F (**2**) and confirmed the sulfate groups attachment to C-4 Glc4 and C-6 Glc4 in **2**. The deshielding of the signal of C-4 Glc4 to δ_C_ 75.5 due to the *α*-shifting effect of the sulfate group in the spectrum of **2** compared with the same signal in the spectrum of **1** (δ_C_ 69.5) and corroborated the position of the sulfate group. Therefore, chilensoside F (**2**) has a tetrasulfated sugar chain with two disulfated glucose residues.

The (*−*)ESI-MS/MS of **2** (Appendix A) demonstrated the fragmentation of [M_4Na_–2Na]^2−^ ion at *m*/*z* 804.2 that led to the ion peak at *m*/*z* 533.7 [M_4Na_–2Na–MeGlc–GlcSO_3_Na+H]^2−^ and fragmentation of [M_4Na_–3Na]^3−^ ion at *m*/*z* 528.5 resulted in the ion peaks at *m*/*z* 488.8 [M_4Na_–3Na–HSO_4_Na]^3−^ and 430.1 [M_4Na_–3Na–MeGlc–SO_4_Na+H]^3−^.

All these data indicate that chilensoside F (**2**) is 3*β*-*O*-{3,6-*O*-sodium disulfate-*β*-D-glucopyranosyl-(1→4)-*β*-D-quinovopyranosyl-(1→2)-[3-*O*-methyl-*β*-D-glucopyranosyl-(1→3)-4,6-*O*-sodium disulfate-*β*-D-glucopyranosyl-(1→4)]-*β*-D-xylopyranosyl}-16-oxoholosta-9(11),24(25)-diene.

The molecular formula of chilensoside G (**3**) was determined to be C_66_H_100_O_44_S_4_Na_4_ from the [M_4Na_–2Na]^2−^ ion peak at *m*/*z* 885.2161 (calc. 885.2138), [M_4Na_–3Na]^3−^ ion peak at *m*/*z* 582.4816 (calc. 582.4795), and [M_4Na_–4Na]^4−^ ion peak at *m*/*z* 431.1139 (calc. 431.1123) in the (*−*)HR-ESI-MS (Appendix A).

The ^1^H and ^13^C NMR spectra of the carbohydrate chain of chilensoside G (**3**) (Table 4, Appendix A) demonstrated six characteristic doublets of anomeric protons at δ_H_ 4.65–5.38 (*J* = 6.8–8.7 Hz) and six signals of anomeric carbons at δ_C_ 102.3–104.7 that corresponded to hexasaccharide chain having *β*-configurations of glycosidic bonds. The extensive analysis of the ^1^H,^1^H-COSY, 1D TOCSY, HSQC, ROESY, and HMBC spectra of **3** indicated the presence of one xylose (Xyl1), one quinovose (Qui2), three glucose (Glc3, Glc4, Glc5), and one 3-*O*-methylglucose (MeGlc6) residue. The monosaccharides were connected to each other and to the aglycone by the glycosidic linkages located at typical sea cucumber glycoside positions, which was confirmed by the correlations in the ROESY and HMBC spectra of **3**: H-1 Xyl1/H-3 (C-3) of the aglycone, H-1 Qui2/H-2 (C-2) Xyl1, H-1 Glc3/H-4 (C-4) Qui2, H-1 Glc4/H-3 (C-3) Glc3, H-1 Glc5/H-4 (C-4) Xyl1, and H-1 MeGlc6/H-3 (C-3) Glc5 (Table 4, Appendix A).

The positions of sulfate groups were established as a result of NMR spectra analyses. The typical values of chemical shifts observed due to the sulfate groups shifting effects were found. The signal of C-6 Glc3 was deshielded to δ_C_ 67.4, and the signal of C-5 Glc3—shielded to δ_C_ 74.7, indicating the sulfation of the hydroxymethylene group of this residue. At the same time, the signal of C-3 Glc3 was observed at δ_C_ 86.1 due to the glycosylation effect appearing because of the attachment of the terminal (Glc4) unit to this position. The ROESY correlation H-1 Glc4/H-3 Glc3 confirmed this supposition. As a result of the analysis of the isolated spin system corresponding to the Glc5 residue, two deshielded signals (in comparison with the signals of non-sulfated carbons) were assigned to sulfated carbons, C-4 Glc5 (δ_C_ 75.3) and C-6 Glc5 (δ_C_ 68.2). Another downfield shifted signal at δ_C_ 80.1 was attributed to position 3 of Glc5 glycosylated by the sixth monosaccharide unit. The last sulfate group was attached to C-6 of the terminal residue in the upper semi-chain—MeGlc6. A corresponding signal (C-6 MeGlc6) was observed at δ_C_ 66.8. The presence of a terminal non-methylated glucose unit (Glc4) was established by the shielded signal of C-3 Glc4 at δ_C_ 77.2, which was observed instead of the signal at δ_C_ ~86.5 in 3-*O*-methylated derivatives. The absence of the signal of the second OMe-group at δ_C_ ~60.5 in the ^13^C NMR spectrum of **3** was an additional confirmation. Hence, chilensoside G (**3**) is a tetrasulfated hexaoside that expanded the list of the most polar glycosides of sea cucumbers found so far.

The (*−*)ESI-MS/MS of **3** (Appendix A) demonstrated the fragmentation of [M_4Na_–2Na]^2−^ ion with *m*/*z* 885.2 resulted in the ion-peaks at *m*/*z* 825.7 [M_4Na_–2Na–SO_4_Na]^2−^, 746.7 [M_4Na_–2Na–MeGlcSO_3_Na]^2−^, 695.7 [M_4Na_–2Na–MeGlcSO_3_Na–SO_3_Na]^2−^, and 614.7 [M_4Na_–2Na–MeGlcSO_3_Na–Glc–SO_3_Na]^2−^. The fragmentation of [M_4Na_–3Na]^3−^ ion at *m*/*z* 582.5 led to the presence of the ion peaks at *m*/*z* 548.8 [M_4Na_–3Na–SO_3_Na]^3−^, 542.8 [M_4Na_–3Na–NaHSO_4_]^3−^, and 484.1 [M_4Na_–3Na–MeGlcSO_3_Na]^3−^.

These data indicate that chilensoside G (**3**) is 3*β*-*O*-{*β*-D-glucopyranosyl-(1→3)-6-*O*-sodium sulfate-*β*-D-glucopyranosyl-(1→4)-*β*-D-quinovopyranosyl-(1→2)-[6-*O*-sodium sulfate-3-*O*-methyl-*β*-D-glucopyranosyl-(1→3)-4,6-*O*-sodium disulfate-*β*-D-glucopyranosyl-(1→4)]-*β*-D-xylopyranosyl}-16-oxoholosta-9(11),24(25)-diene.

### 2.2. Bioactivity of the Glycosides

The cytotoxic activities of chilensosides E–G (**1**–**3**) against human cell lines, erythrocytes, and cancer cells, including neuroblastoma SH-SY5Y, adenocarcinoma HeLa, colorectal adenocarcinoma DLD-1, leukemia promyeloblast HL-60, and monocytic THP-1 have been studied. The earlier tested chitonoidoside L [3] was used as the positive control in all the tests (Table 5). The activity of the glycosides against SH-SY5Y, HeLa, and DLD-1 cells was examined using an MTT assay and against HL-60 and THP-1 cells using an MTS assay.

Erythrocytes, in agreement with earlier published data [15,19], exhibited an increased sensitivity to the membranolytic action of sea cucumber glycosides compared to cancer cells. The erythrocytes are a traditional and convenient model for investigating the membranolytic action of the glycosides. All compounds **1**–**3** showed moderate hemolytic activity, allowing us to suppose the cytotoxic doses of the investigated compounds will be relatively high. Chilensosides E–G (**1**–**3**) were not cytotoxic against cancer cell lines even at the maximal tested concentration (100 µM). As previously observed, the presence of four sulfate groups alone did not deplete the activity [17]. However, the decreasing membranolytic activity caused by the increasing number of sulfates was reported earlier for some of the glycosides [20]. Generally, the influence of sulfate groups on the activity of the glycosides significantly depends on the structural characteristics of their carbohydrate chains [9]. So, in the case of chilensosides E–G (**1**–**3**), the combination of sulfates quantity and positions presumably negatively affected their activity. This was in good agreement with the previous tests of cytotoxicity of tetrasulfated chilensoside D, which has been inactive in relation to three of five cancer cell lines [15]. The analysis of the structure-activity relationships of the glycosides isolated from *P. chilensis* and the comparison between the effects of trisulfated chilensoside C, tetrasulfated chilensosides D [15], and E–G (**1**–**3**) have allowed the finding of the common feature of tetrasulfated glycosides that presumably extremely decreased their activity against cancer cells—the presence of a sulfate group at C-6 Glc3 (on the bottom semi-chain). However, the much more complicated influence of the glycosides’ different structural features, including aglycones’ structures, carbohydrate chains, architecture and composition, and their combinations on the glycoside/membrane interactions is obvious.

### 2.3. Biogenesis of Chilensosides A–G

The majority of chilensosides share the same overall structures, including the same aglycones and monosaccharide composition. Consequently, biosynthesis of the glycosides of *P. chilensis* looks somewhat strictly directed. However, the combinatorial features of biosynthesis [18,21] have become evident when the character of sulfation of sugar chains of these glycosides is analyzed. The biogenetic analysis of this series of glycosides has revealed that they form a network instead of biogenetic rows due to the enzymatic introduction of sulfate groups in almost all possible positions (Figure 3). The trend has been derived from the biogenetic analysis that the upper semi-chains are sulfated before the bottom ones, and C-6 of the glucose residues attached to C-4 Xyl1 take precedence in the sulfation.

Noticeably, the presence of a sulfate group at C-3 Glc3—the terminal monosaccharide residue in the bottom semi-chain of compounds **1** and **2**—excludes the possibility of further sugar chain elongation. Thus, chilensosides C [15] and F (**2**) could not be the biosynthetic precursors of hexaoside chilensoside G (**3**). More likely, chilensoside G (**3**) is biosynthesized through the chilensosides A and D (Figure 2), having a free hydroxyl group available for glycosylation. Presumably, the processes of glycosylation and sulfation are concurrent biosynthetic stages. They can be shifted in time in relation to each other, a characteristic feature of the mosaic type of biosynthesis [18,21].

## 3. Materials and Methods

### 3.1. General Experimental Procedures

Specific rotation was measured on a PerkinElmer 343 Polarimeter (PerkinElmer, Waltham, MA, USA); NMR spectra were registered on a Bruker AMX 500 (Bruker BioSpin GmbH, Rheinstetten, Germany) (500.12/125.67 MHz (^1^Н/^13^C) spectrometer; ESI MS (positive and negative ion modes) spectra were registered on an Agilent 6510 Q-TOF apparatus (Agilent Technology, Santa Clara, CA, USA), sample concentration 0.01 mg/mL; HPLC was conducted on an Agilent 1260 Infinity II equipped with a differential refractometer (Agilent Technology, Santa Clara, CA, USA); columns were used: Supelcosil LC-Si (4.6 *×* 150 mm, 5 µm) and Discovery Ascentis RP-Amide (10 *×* 250 mm, 5 µm) (Supelco, Bellefonte, PA, USA), Diasfer 110 C-8 (4.6 *×* 250 mm, 5 µm) (Biochemmack, Moscow, Russia).

### 3.2. Animals and Cells

The sea cucumber *Paracaudina chilensis* (family Caudinidae; order Molpadida) (36 specimens) was harvested in Troitsa Bay, Sea of Japan, in August 2019 by scuba diving 2–5 m in depth. The taxonomic position of the animals was determined by Boris B. Grebnev. The voucher specimen PIBOC-2019-MES-0135 is kept in G.B. Elyakov PIBOC FEB RAS, Vladivostok, Russia.

Human erythrocytes were purchased from the Station of Blood Transfusion in Vladivostok. The cells of the human adenocarcinoma line (HeLa) were provided by the N.N. Blokhin National Medicinal Research Center of Oncology of the Ministry of Health Care of the Russian Federation (Moscow, Russia). The human colorectal adenocarcinoma line DLD-1 CCL-221™ cells, human monocytic THP-1 TIB-202^™^ cells, human neuroblastoma line SH-SY5Y CRL-2266^™^, and human promyeloblast cell line HL-60 CCL-240 were received from ATCC (Manassas, VA, USA). The HeLa cell line was cultured in the medium of DMEM (Gibco Dulbecco’s Modified Eagle Medium) with 1% penicillin /streptomycin sulfate (Biolot, St. Petersburg, Russia) and 10% fetal bovine serum (FBS) (Biolot, St. Petersburg, Russia). The cells of THP-1, HL-60, and DLD-1 lines were cultured in the medium of RPMI with 1% penicillin/streptomycin (Biolot, St. Petersburg, Russia) and 10% fetal bovine serum (FBS) (Biolot, St. Petersburg, Russia). The cells were incubated at 37 °C in a humidified atmosphere with 5% (*v*/*v*) CO_2_. SH-SY5Y cells were cultured in MEM (Minimum Essential Medium) with 1% penicillin/streptomycin sulfate (Biolot, St. Petersburg, Russia) and with fetal bovine serum (Biolot, St. Petersburg, Russia) to a final concentration of 10%.

The study was carried out in accordance with the guidelines of the Declaration of Helsinki and approved by the Ethics Committee of G.B > Elyakov Pacific Institute of Bioorganic Chemistry (Protocol No. 0037.12.03.2021).

### 3.3. Extraction and Isolation

The ethanol extract of the sea cucumbers was purified by standard methodology [15,18], including column hydrophobic and Si gel chromatography. For the latter stage, the stepwise gradient of solvent systems CHCl3/EtOH/H2O: 100:100:17 → 100:125:25 → 100:150:50 as mobile phase was applied, followed by the additional purification of the obtained fractions with CHCl3/EtOH/H2O (100:125:25) as the mobile phase. Finally, the subfractions: I.0 (22 mg), I.1 (120 mg), II (286 mg), III.1 (66 mg), and III.2 (177 mg) were isolated [15]. HPLC of the subfraction II on the silica-based column Supelcosil LC-Si (4.6 × 150 mm, 5 µm) with CHCl_3_/MeOH/H_2_O (55/30/4) as the mobile phase resulted in the isolation of two fractions (II.1 and II.2). The subsequent HPLC of fraction II.2 on the Supelco Ascentis RP-Amide (10 × 250 mm) column with MeOH/H_2_O/NH_4_OAc (1 M water solution), ratio (60/38.5/1.5), as the mobile phase led to the isolation of four subfractions. The re-chromatography of two of them on a Diasfer 110 C-8 (4.6 × 250 mm) column with MeOH/H_2_O/NH_4_OAc (1 M water solution) (50/48/2) as the mobile phase applied for the separation of each subfraction, resulted in the isolation of chilensosides E (**1**) (2.2 mg, R_t_ 14.12 min) and F (**2**) (2.8 mg, R_t_ 17.25 min). The subfraction III.2 was submitted to HPLC on a Supelco Ascentis RP-Amide (10 × 250 mm) column with CH_3_CN/H_2_O/NH_4_OAc (1 M water solution), ratio (30/68/2), as the mobile phase to give two main fractions and some minor ones. The repeated HPLC of one of the main fractions in the same conditions led to the isolation of 11.0 mg of chilensoside G (**3**) (R_t_ 16.67 min).

#### 3.3.1. Chilensoside E (**1**)

Colorless powder; [α]_D_^20^–39° (*c* 0.1, H_2_O). NMR: Appendix A and Table 1, Appendix A. (*−*)HR-ESI-MS *m*/*z*: 804.1857 (calc. 804.1874) [M_4Na_–2Na]^2−^, 528.4622 (calc. 528.4619) [M_4Na_–3Na]^3−^, 390.5999 (calc. 390.5991) [M_4Na_–4Na]^4−^); (*−*)ESI-MS/MS *m*/*z*: 665.7 [M_4Na_–2Na–C_7_H_11_O_8_SNa (MeGlcSO_3_Na)+H]^2−^, 456.8 [M_4Na_–3Na–HSO_4_Na–SO_3_Na+2H]^3−^, 430.1 [M_4Na_–3Na–C_7_H_12_O_9_SNa (MeGlcSO_3_Na)]^3−^.

#### 3.3.2. Chilensoside F (**2**)

Colorless powder; [α]_D_^20^–42° (*c* 0.1, H_2_O). NMR: Appendix A and Table 2, Appendix A. (*−*)HR-ESI-MS *m*/*z*: 804.1875 (calc. 804.1874) [M_4Na_–2Na]^2−^, 528.4628 (calc. 528.4619) [M_4Na_–3Na]^3−^, 390.5998 (calc. 390.5991) [M_4Na_–4Na]^4−^); (*−*)ESI-MS/MS *m*/*z*: 533.7 [M_4Na_–2Na–C_7_H_13_O_5_(MeGlc)–C_6_H_9_O_11_S_2_Na_2_(Glc(SO_3_Na)_2_+H]^2−^, 488.8 [M_4Na_–3Na–HSO_4_Na]^3−^, 430.1 [M_4Na_–3Na–C_7_H_13_O_5_ (MeGlc)–SO_4_Na+H]^3−^.

#### 3.3.3. Chilensoside G (**3**)

Colorless powder; [α]_D_^20^–53° (*c* 0.1, H_2_O). NMR: Appendix A and Table 3, Appendix A. (*−*)HR-ESI-MS *m*/*z*: 885.2161 (calc. 885.2138) [M_4Na_–2Na]^2−^, 582.4816 (calc. 582.4795) [M_4Na_–3Na]^3−^, 431.1139 (calc. 431.1123) [M_4Na_–4Na]^4−^; (*−*)ESI-MS/MS *m*/*z*: 825.7 [M_4Na_–2Na–SO_4_Na]^2−^, 746.7 [M_4Na_–2Na–C_7_H_12_O_8_SNa(MeGlcSO_3_Na)]^2−^, 695.7 [M_4Na_–2Na–C_7_H_12_O_8_SNa(MeGlcSO_3_Na)–SO_3_Na+H]^2−^, 614.7 [M_4Na_–2Na–C_7_H_12_O_8_SNa (MeGlcSO_3_Na)–C_6_H_11_O_5_(Glc)–SO_3_Na]^2−^_,_ 548.8 [M_4Na_–3Na–SO_3_Na]^3−^, 542.8 [M_4Na_–3Na–NaHSO_4_]^3−^, 484.1 [M_4Na_–3Na–C_7_H_12_O_8_SNa (MeGlcSO_3_Na)]^3−^.

### 3.4. Cytotoxic Activity (MTT Assay) (for SH-SY5Y, HeLa, and DLD-1 Cells)

All substances (including chitonoidoside L used as a positive control) were tested in concentrations from 0.1 µM to 100 µM using a 2-fold dilution in d-H2O. The cell suspension (180 µL) and solutions (20 µL) of tested glycosides in different concentrations were injected in wells of 96-well plates (SH-SY5Y, 1×104 cells/well, HeLa and DLD-1, 6×10^3^/200 µL) and incubated at 37 °C for 24 h in the atmosphere with 5% CO_2_. After the incubation, the glycosides with the medium were replaced by 100 µL of fresh medium. Then, 10 µL of MTT (3-(4,5-dimethylthiazol-2-yl)-2,5-diphenyltetrazolium bromide) (Sigma-Aldrich, St. Louis, MO, USA) stock solution (5 mg/mL) were added to each well, followed by incubation of the microplate for 4 h. After this procedure, 100 µL of SDS-HCl solution (1 g SDS/10 mL d-H2O/17 µL 6 N HCl) was added to each well and incubated for 18 h. The absorbance of the converted dye formazan was determined with a Multiskan FC microplate photometer (Thermo Fisher Scientific, Waltham, MA, USA) at 570 nm. Cytotoxic activity of the tested glycosides was calculated as a concentration that caused 50% cell metabolic activity inhibition (IC50). The experiments were conducted in triplicate, *p* < 0.05.

### 3.5. Cytotoxic Activity (MTS Assay) (for HL-60 and THP-1 Cells)

The cells of THP-1 (6 × 10^3^/200 µL) and HL-60 line (10 × 10^3^/200 µL) were placed in 96-well plates at 37 °C for 24 h in a 5% CO_2_ incubator. The cells were treated with tested glycosides and chitonoidoside L as a positive control at concentrations between 0 and 100 µM for an additional 24 h. Then the cells were incubated with 10 µL MTS ([3-(4,5-dimethylthiazol-2-yl)-5-(3-carboxymethoxyphenyl)-2-(4-sulfophenyl)-2H-tetrazolium) for 4 h, and the absorbance in each well was determined at 490/630 nm with a plate reader PHERA star FS (BMG Labtech, Ortenberg, Germany). The experiments were conducted in triplicate. The results were presented as the percentage of inhibition that produced a reduction in absorbance after tested glycoside treatment compared to the non-treated cells (negative control), *p* < 0.01.

### 3.6. Hemolytic Activity

Erythrocytes were obtained from human blood (AB(IV) Rh+) by centrifugation with phosphate-buffered saline (PBS) (pH 7.4) at 4 °C for 5 min three times at 450 g on a centrifuge LABOFUGE 400R (Heraeus, Hanau, Germany). Then, the erythrocytes residue was resuspended in ice-cold phosphate saline buffer (pH 7.4) to a final optical density of 1.5 at 700 nm and kept on ice [22]. For the hemolytic assay, 180 µL of erythrocyte suspension was mixed with 20 µL of test compound solution (including chitonoidoside L used as a positive control) in V-bottom 96-well plates. After 1 h of incubation at 37 °C, the plates were exposed to centrifugation for 10 min at 900 g in a laboratory centrifuge LMC-3000 (Biosan, Riga, Latvia) [22]. Then, 100 µL of supernatant was carefully decanted and transferred into new flat plates. The erythrocyte lysis values were measured by measuring the supernatant’s hemoglobin concentration with a microplate photometer Multiskan FC (Thermo Fisher Scientific, Waltham, MA, USA), λ = 570 nm [23]. The effective dose causing 50% hemolysis of erythrocytes (ED50) was calculated with the computer program SigmaPlot 10.0. All the experiments were carried out in triple repetitions, *p* < 0.01.

## 4. Conclusions

As a result of the investigation of the glycosidic composition of the sea cucumber *Paracaudina chilensis*, taking into account this paper and a previously published paper [15], the structures of eight new glycosides, chilensosides A–G have been established, and their cytotoxic activity has been studied. The structural diversity of these compounds concerned the sugar moieties, mainly the number and positions of sulfate groups. Seven glycosides shared pentasaccharides branched by C-4 Xyl1 chains with the same monosaccharide composition and sequence of residues. One glycoside from the series—chilensoside G (**3**)—contained six monosaccharide residues having additional glucose units in the bottom semi-chain. Analogously, seven glycosides have identical aglycones, and only one, chilensoside A [15], is characterized by the other structure of the aglycone side chain. Thus, the majority of chilensosides share the same overall structure. They are formed as a result of the same cascade of enzymatic reactions making this biosynthetic pathway predominant and minimizing the mosaicism (combinatorial character) [18] of the glycosides biosynthesis. However, the combinatorial features become evident in the mode of sulfation of sugar chains of these glycosides because of the attachment of sulfate groups in somewhat different positions.

## Figures and Tables

**Figure 1 marinedrugs-21-00114-f001:**
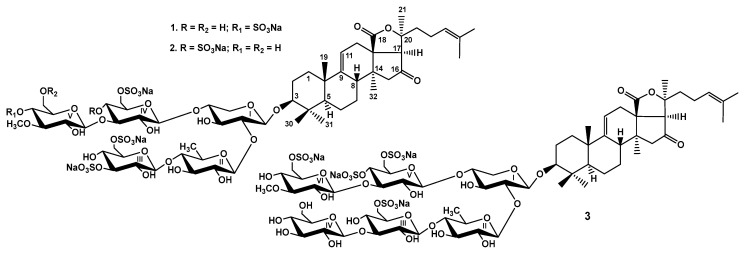
Chemical structures of glycosides isolated from *Paracaudina chilensis*: **1**—chilensoside E; **2**—chilensoside F; **3**—chilensoside G.

**Figure 2 marinedrugs-21-00114-f002:**
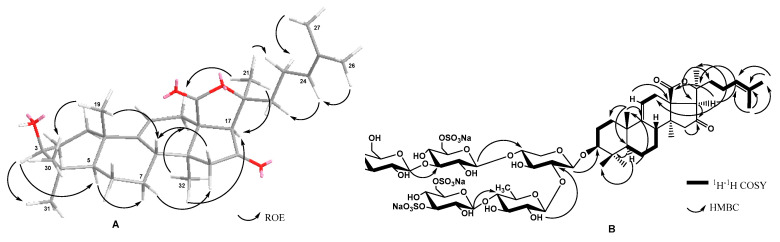
Key ROE- (**A**), HMBC and COSY (**B**) correlations of chilensoside E. The oxygen atoms have marked with red color.

**Figure 3 marinedrugs-21-00114-f003:**
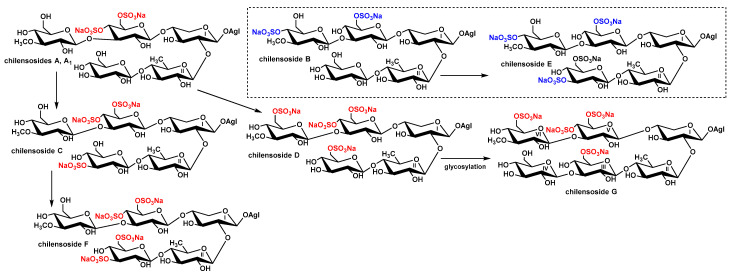
The directions of sulfation in the process of biosynthesis of the glycosides of *P. chilensis*. Uncombined biogenetic rows are highlighted by blue and red colors.

**Table 1 marinedrugs-21-00114-t001:** ^13^C and ^1^H NMR chemical shifts, HMBC and ROESY correlations of aglycone moiety of chilensoside E (**1**).

Position	δ_C_ Mult. *^a^*	δ_H_ Mult. (*J* in Hz) *^b^*	HMBC	ROESY
1	36.0 CH_2_	1.73 m		H-11
		1.30 m		H-3
2	26.8 CH_2_	2.06 m		
		1.84 m		H-19, H-30
3	88.4 CH	3.11 dd (4.7; 11.8)		H-1, H-5, H-31, H1-Xyl1
4	39.5 C			
5	52.7 CH	0.79 brd (11.8)	C: 4, 19, 30	H-1, H-3, H-7
6	20.9 CH_2_	1.59 m		
7	28.3 CH_2_	1.60 m		H-15
		1.12 m		
8	38.6 CH	3.14 m		H-6
9	151.1 C			
10	39.5 C			
11	111.3 CH	5.27 brs	C: 10, 13	H-1
12	31.9 CH_2_	2.64 brd (16.5)	C: 11, 18	H-17
		2.48 dd (5.9; 16.5)	C: 11, 14	
13	56.0 C			
14	42.0 C			
15	51.8 CH_2_	2.42 d (16.0)	C: 13, 16, 17, 32	
		2.11 d (16.0)	C: 14, 16, 32	H-8
16	214.6 C			
17	61.8 CH	2.89 s	C: 12, 13, 16, 18, 20, 21	H-12, H-23, H-32
18	176.9 C			
19	21.9 CH_3_	1.27 s	C: 1, 5, 9, 10	H-1, H-2, H-8, H-30
20	83.1 C			
21	26.6 CH_3_	1.48 s	C: 17, 20, 22	H-12, H-17, H-23
22	38.6 CH_2_	1.80 m		
		1.59 m		
23	23.0 CH_2_	2.27 m		
		2.08 m		
24	124.1 CH	5.03 m		H-22
25	132.1 C			
26	25.5 CH_3_	1.56 s	C: 24, 25, 27	H-24
27	17.4 CH_3_	1.54 s	C: 24, 25, 26	H-23
30	16.4 CH_3_	0.89 s	C: 3, 4, 5, 31	H-2, H-6, H-19, H-31
31	27.8 CH_3_	1.10 s	C: 3, 4, 5, 30	H-3, H-5, H-6, H-30
32	20.5 CH_3_	0.89 s	C: 8, 13, 14, 15	H-7, H-12, H-15, H-17

*^a^* Recorded at 176.04 MHz in C_5_D_5_N/D_2_O (4/1). *^b^* Recorded at 700.13 MHz in C_5_D_5_N/D_2_O (4/1). The original spectra of **1** are provided in Appendix A.

**Table 2 marinedrugs-21-00114-t002:** ^13^C and ^1^H NMR chemical shifts, HMBC and ROESY correlations of the carbohydrate moiety of chilensoside E (**1**).

Atom	δ_C_ Mult. ^*a*,*b*,*c*^	δ_H_ Mult. (*J* in Hz) *^d^*	HMBC	ROESY
Xyl1 (1→C-3)				
1	104.6 CH	4.61 d (7.4)	C: 3	H-3; H-5 Xyl1
2	**83.4** CH	3.70 m	C: 3 Xyl1	H-1 Qui2
3	75.0 CH	3.93 m		
4	**80.8** CH	3.94 m	C: 3 Xyl1; 1 Glc4	H-1 Glc4; H-2 Xyl1
5	63.4 CH_2_	4.33 dd (6.0; 12.3)		
		3.58 m		H-1 Xyl1
Qui2 (1→2Xyl1)				
1	104.6 CH	4.74 d (8.3)	C: 2 Xyl1	H-2 Xyl1; H-3, 5 Qui2
2	75.5 CH	3.92 t (9.4)	C: 3 Qui2	H-4 Qui2
3	74.4 CH	4.06 t (9.4)		H-1, 5 Qui2
4	**86.0** CH	3.30 t (9.4)	C: 1 Glc3	H-1 Glc3
5	71.7 CH	3.60 dd (6.0; 9.4)		H-1, 3 Qui2
6	17.4 CH_3_	1.54 d (6.0)	C: 4, 5 Qui2	H-4 Qui2
Glc3 (1→4Qui2)				
1	104.1 CH	4.67 d (7.9)	C: 4 Qui2	H-4 Qui2; H-3, 5 Glc3
2	72.8 CH	3.81 t (7.9)	C: 1, 3 Glc3	
3	*83.8* CH	4.98 t (7.9)		H-1, 5 Glc3
4	69.5 CH	3.80 t (7.9)	C: 3, 5 Glc3	H-6 Glc3
5	74.2 CH	4.07 t (7.9)		
6	*67.4* CH_2_	4.91 brd (9.8)		
		4.52 dd (7.3; 11.0)		
Glc4 (1→4Xyl1)				
1	103.4 CH	4.85 d (7.9)	C: 4 Xyl1	H-4 Xyl1; H-3, 5 Glc4
2	74.0 CH	3.83 t (7.9)		
3	**85.8** CH	4.17 t (9.2)	C: 1 MeGlc5	H-1 MeGlc5; H-1 Glc4
4	69.5 CH	3.68 t (9.2)	C: 5 Glc4	
5	74.4 CH	4.16 m		
6	*67.7* CH_2_	4.95 brd (10.5)		
		4.44 brdd (7.2; 11.2)		
MeGlc5 (1→3Glc4)				
1	104.3 CH	5.16 d (8.3)	C: 3 Glc4	H-3 Glc4; H-5 MeGlc5
2	74.0 CH	3.87 t (8.3)	C: 1 MeGlc5	H-4 MeGlc5
3	85.3 CH	3.71 t (8.3)	C: 4 MeGlc5; OMe	H-1 Me Glc5
4	*76.1* CH	4.88 t (8.8)	C: 5 MeGlc5	
5	76.4 CH	3.86 m		H-1 MeGlc5
6	61.7 CH_2_	4.49 d (11.6)		
		4.33 m		
OMe	60.8 CH_3_	3.93 s	C: 3 MeGlc5	H-3 MeGlc5

*^a^* Recorded at 125.67 MHz in C_5_D_5_N/D_2_O (4/1). *^b^* Bold = interglycosidic positions. *^c^* Italic = sulfate position. *^d^* Recorded at 500.12 MHz in C_5_D_5_N/D_2_O (4/1). Multiplicity by 1D TOCSY. The original spectra of **1** are provided in Appendix A.

**Table 3 marinedrugs-21-00114-t003:** ^13^C and ^1^H NMR chemical shifts, HMBC and ROESY correlations of the carbohydrate moiety of chilensoside F (**2**).

Atom	δ_C_ Mult. ^*a*,*b*,*c*^	δ_H_ Mult. (*J* in Hz) *^d^*	HMBC	ROESY
Xyl1 (1→C-3)				
1	104.6 CH	4.62 d (7.7)	C: 3	H-3; H-3, 5 Xyl1
2	**82.9** CH	3.71 t (7.7)	C: 1, 3 Xyl1	H-1 Qui2
3	75.1 CH	3.93 t (7.7)	C: 2, 4 Qui2	H-1 Xyl1
4	**81.2** CH	3.92 m	C: 1 Glc4	H-1 Glc4
5	63.4 CH_2_	4.28 dd (5.1; 11.5)	C: 3 Xyl1	
		3.57 m		H-1, 3 Xyl1
Qui2 (1→2Xyl1)				
1	104.7 CH	4.75 d (7.3)	C: 2 Xyl1	H-2 Xyl1; H-3, 5 Qui2
2	75.3 CH	3.94 t (9.3)		H-4 Qui2
3	74.4 CH	4.06 t (9.3)	C: 2, 4 Qui2	H-1, 5 Qui2
4	**85.9** CH	3.31 t (9.3)	C: 1 Glc3; 5 Qui2	H-1 Glc3; H-2 Qui2
5	71.7 CH	3.60 m		H-1, 3 Qui2
6	17.4 CH_3_	1.53 d (6.4)	C: 4, 5 Qui2	H-4 Qui2
Glc3 (1→4Qui2)				
1	104.1 CH	4.68 d (7.5)	C: 4 Qui2	H-4 Qui2; H-3, 5 Glc3
2	72.8 CH	3.80 t (9.0)	C: 1, 3 Glc3	
3	*83.9* CH	4.98 t (9.0)	C: 2, 4 Glc3	H-1, 5 Glc3
4	69.5 CH	3.82 t (9.0)	C: 3, 5, 6 Glc3	
5	74.3 CH	4.08 t (9.0)		H-1, 3 Glc3
6	*67.4* CH_2_	4.92 brd (11.7)		
		4.53 dd (7.6; 11.7)	C: 5 Glc3	H-4 Glc3
Glc4 (1→4Xyl1)				
1	103.5 CH	4.81 d (7.8)	C: 4 Xyl1	H-4 Xyl1; H-3, 5 Glc4
2	74.0 CH	3.92 t (8.4)	C: 1, 3 Glc4	
3	**82.8** CH	4.37 t (8.4)	C: 1 MeGlc5; 2, 4 Glc4	H-1 MeGlc5
4	*75.5* CH	4.74 t (8.4)	C: 5, 6 Glc4	H-2 Glc4
5	73.8 CH	4.33 m		H-1 Glc4
6	*68.5* CH_2_	5.50 m		
		4.62 t (10.9)		H-4 Glc4
MeGlc5 (1→3Glc4)				
1	104.5 CH	5.20 d (8.2)	C: 3 Glc4	H-3 Glc4; H-3, 5 MeGlc5
2	74.5 CH	4.00 t (9.4)	C: 1, 3 MeGlc5	
3	86.9 CH	3.65 t (9.4)	C: 2, 4 MeGlc5; OMe	H-1, 5 Me Glc5; OMe
4	70.0 CH	3.91 t (9.4)	C: 3, 5, 6 MeGlc5	H-6 MeGlc5
5	77.5 CH	3.87 m		H-1 MeGlc5
6	62.0 CH_2_	4.34 brd (11.7)		
		4.09 dd (7.0; 11.7)	C: 5 MeGlc5	
OMe	60.3 CH_3_	3.76 s	C: 3 MeGlc5	

*^a^* Recorded at 125.67 MHz in C_5_D_5_N/D_2_O (4/1). *^b^* Bold = interglycosidic positions. *^c^* Italic = sulfate position. *^d^* Recorded at 500.12 MHz in C_5_D_5_N/D_2_O (4/1). Multiplicity by 1D TOCSY. The original spectra of **2** are provided in Appendix A.

**Table 4 marinedrugs-21-00114-t004:** ^13^C and ^1^H NMR chemical shifts, HMBC and ROESY correlations of the carbohydrate moiety of chilensoside G (**3**).

Atom	δ_C_ Mult. ^*a*,*b*,*c*^	δ_H_ Mult. (*J* in Hz) *^d^*	HMBC	ROESY
Xyl1 (1→C-3)				
1	104.7 CH	4.65 d (6.8)	C: 3	H-3; H-3, 5 Xyl1
2	**82.2** CH	3.85 t (8.7)	C: 1 Xyl1	H-1 Qui2
3	75.0 CH	4.10 t (8.7)	C: 2, 4 Qui2	H-5 Xyl1
4	**79.5** CH	4.05 m		H-1 Glc5
5	63.4 CH_2_	4.34 dd (4.9; 11.7)	C: 3 Xyl1	
		3.63 brdd (8.7; 11.7)		H-1 Xyl1
Qui2 (1→2Xyl1)				
1	104.5 CH	4.89 d (8.0)	C: 2 Xyl1	H-2 Xyl1; H-3, 5 Qui2
2	75.4 CH	3.83 t (8.9)	C: 1, 3 Qui2	H-4 Qui2
3	74.7 CH	3.92 t (8.3)	C: 2 Qui2	H-1 Qui2
4	**86.9** CH	3.34 t (8.3)	C: 1 Glc3; 3, 5 Qui2	H-1 Glc3; H-2 Qui2
5	71.4 CH	3.61 dd (6.2; 8.3)		H-1, 3 Qui2
6	17.6 CH_3_	1.57 d (6.5)	C: 4, 5 Qui2	H-4 Qui2
Glc3 (1→4Qui2)				
1	104.2 CH	4.71 d (7.8)	C: 4 Qui2	H-4 Qui2; H-3, 5 Glc3
2	73.4 CH	3.83 t (8.6)	C: 1, 3 Glc3	
3	**86.1** CH	4.16 t (8.6)	C: 1 Glc4; 2, 4 Glc3	H-1 Glc4; H-1 Glc3
4	69.3 CH	3.75 t (8.6)	C: 3, 5, 6 Glc3	H-6 Glc3
5	74.7 CH	4.08 t (8.6)		H-1 Glc3
6	*67.4* CH_2_	4.95 brd (10.2)	C: 4 Glc3	
		4.54 dd (7.1; 11.0)		H-4 Glc3
Glc4 (1→3Glc3)				
1	104.6 CH	5.19 d (7.8)	C: 3 Glc3	H-3 Glc3; H-3, 5 Glc4
2	74.9 CH	3.91 t (8.6)	C: 3 Glc4	
3	77.2 CH	4.09 t (8.6)	C: 2, 4 Glc4	H-1 Glc4
4	71.0 CH	3.88 t (8.6)		
5	77.7 CH	3.90 t (8.6)		H-1 Glc4
6	61.9 CH_2_	4.35 d (11.8)		
		4.05 dd (5.5; 11.8)	C: 5 Glc4	
Glc5 (1→4Xyl1)				
1	102.7 CH	4.81 d (8.7)	C: 4 Xyl1	H-4 Xyl1; H-3, 5 Glc5
2	73.5 CH	3.96 t (8.7)	C: 1, 3 Glc5	
3	**80.1** CH	4.52 t (8.7)	C: 1 MeGlc6; 2, 4 Glc5	H-1 MeGlc6; H-1, 5 Glc5
4	*75.3* CH	4.76 t (8.7)	C: 3, 5, 6 Glc5	H-6 Glc5
5	73.8 CH	4.25 t (8.7)	C: 4, 6 Glc5	H-1, 3 Glc5
6	*68.2* CH_2_	5.39 d (10.9)		
		4.66 dd (7.6; 10.9)	C: 5 MeGlc5	
MeGlc6 (1→3Glc5)				
1	102.3 CH	5.38 d (7.8)	C: 3 Glc5	H-3 Glc5; H-3, 5 MeGlc6
2	74.0 CH	3.94 t (7.8)	C: 1, 3 MeGlc6	
3	86.3 CH	3.61 t (8.9)	OMe; C: 2, 4 MeGlc6	OMe; H-1, 5 MeGlc6
4	69.4 CH	4.04 t (8.9)	C: 3, 5, 6 MeGlc6	
5	75.6 CH	3.97 m		H-1, 3 MeGlc6
6	*66.8* CH_2_	4.94 d (10.4)	C: 4 MeGlc6	
		4.79 dd (5.2; 11.9)		
OMe	60.3 CH_3_	3.73 s	C: 3 MeGlc6	

*^a^* Recorded at 125.67 MHz in C_5_D_5_N/D_2_O (4/1). *^b^* Bold = interglycosidic positions. *^c^* Italic = sulfate position. *^d^* Recorded at 500.12 MHz in C_5_D_5_N/D_2_O (4/1). Multiplicity by 1D TOCSY. The original spectra of **3** are provided in Appendix A.

**Table 5 marinedrugs-21-00114-t005:** The cytotoxic activities of glycosides **1**–**3** and chitonoidoside L (positive control) against human erythrocytes, SH-SY5Y, HeLa, DLD-1, HL-60, and THP-1 human cell lines.

Glycosides	ED_50_, µM, Erythrocytes	Cytotoxicity, IC_50_ µM				
		SH-SY5Y	HeLa	DLD-1	HL-60	THP-1
Chilensoside E (1)	29.89 ± 2.67	>100.0	>100.0	>100.0	>100.0	>100.0
Chilensoside F (2)	34.31 ± 0.63	>100.0	>100.0	>100.0	>100.0	>100.0
Chilensoside G (3)	40.74 ± 1.55	>100.0	>100.0	>100.0	>100.0	>100.0
Chitonoidoside L	1.12 ± 0.10	8.06 ± 0.98	14.36 ± 1.12	9.61 ± 1.24	8.22 ± 0.65	8.32 ± 0.81

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
