# Peer review of "Chilensosides E, F, and G—New Tetrasulfated Triterpene Glycosides from the Sea Cucumber Paracaudina chilensis (Caudinidae, Molpadida): Structures, Activity, and Biogenesis"

_marinedrugs, 2023, doi:10.3390/md21020114_

Round 1

Reviewer 1 Report

This is an elegant paper describing the discovery, structure, biogenesis and bioactivity of three novel tetrasulfated triterpene glycosides from the Sea Cucumber Paracaudina chilensis. These glycosides are named chilensosides E, F, and G. Chilensosides E and F are tetrasulfated pentaosides, while chilensoside G is a tetrasulfated hexaoside. The established structures are based on extensive analysis. The precursor/progeny speculations are all appropriate. These compounds have modest hemolytic effects, but do not have significant effects on the viability of several human cancer cell lines.

The quality of this paper reflects the careful work and analysis that we have come to expect from this group at the Pacific Institute of Bioorganic Chemistry in Vladivostok. This reviewer only has minor comments on the manuscript itself.

1: Generally, the English is good but runs into problems with the use of the past tense in the abstract. Line 15 differ rather than “differed”. Line 17 would read better as: Chilensosides E (1) and F (2) are tetrasulfated pentaosides, and chilensoside G 17 (3) is a tetrasulfated hexaoside.

2: Insert a space between line 179 and 180. At present the table legend runs into the body of the text. This will also solve the next problem, where the heading of table 4 will be pushed onto the next page with the contents of the table.

3: aglycone structures rather than aglycones structures

4: Lines 283, 289, and 295 refer to the chilenosides as “colorless powders. Does this mean white powders of are they translucent?

Author Response

Note: 1: Generally, the English is good but runs into problems with the use of the past tense in the abstract. Line 15 differ rather than “differed”. Line 17 would read better as: Chilensosides E (1) and F (2) are tetrasulfated pentaosides, and chilensoside G 17 (3) is a tetrasulfated hexaoside.

Reply: Fixed

Note: 2: Insert a space between line 179 and 180. At present the table legend runs into the body of the text. This will also solve the next problem, where the heading of table 4 will be pushed onto the next page with the contents of the table.

Reply: All the Table headings are divided from the body text with spaces.

Note: 3:  aglycone structures rather than aglycones structures

Reply: Corrected to: “structures of the aglycones”

Note: 4: Lines 283, 289, and 295 refer to the chilenosides as “colorless powders. Does this mean white powders of are they translucent?

Reply: The color of substance depends on the conditions of drying in vacuo (what solvent was evaporated). For instance, when the glycoside was dissolved in water-containing solvent system (MeOH/H2O, CH3CN/H2O, CHCl3/MeOH/H2O), the following evaporation leads to the formation of white nontransparent powder. However, when the solution of the glycoside in EtOH is evaporated, the translucent cover is formed, as a rule. Additionally, the color depends on the amount of isolated compound. When the amount is more then 10 mg it looks like powder.

Reviewer 2 Report

The authors report the structure elucidaton of three new tetrasulfated triterpene glycosides named chilensosides E, F and G, isolated from the sea cucumber Paracaudina chilensis. Through extensive analysis of 1D and 2D NMR spectra, HR-ESI-MS, and a tentative assignation of the monosaccharide residues as D based on biogenetic analogies with previous known sea cucumber triterpene glycosides, the authors established their structures. In addition, the authors carried out cytotoxic activities, showing moderate hemolytic activity and non-activity against cancer cells. Finally, the authors also discuss about the biosynthesis of the glycosides of P. chilensis.

The main achievement described in the manuscript is the report of three new glycosides from the sea cucumber Paracaudina chilensis. In a previous work from the same group (reference 15), the authors reported the structure of new glycosides, chilensosides A-D, isolated from the same sea cucumber. This work is a continuation of the previous one, and in combination, the structures of chilensosides A-G were established. Although the authors included cytotoxic activities and biogenesis discussion, this is a relatively modest contribution taking into account the previous work. However, I think it is still of interest for the readers of Mar. Drugs and should be published in Mar. Drugs after minor revisions.

The work is thorough and well-presented. The SI is good.

Minor points the authors should modify:

1-Line 230, aglycones, please fix.

2-My main concern is about organization of the paper regarding the biogenesis discussion part. When I saw the title, it includes clearly 3 main points: structure, activity and biogenesis. However, the biogenesis part in the paper is included in the conclusions. I would suggest the authors to move it to a different previous section where the authors focus only in the biosynthesis. I would suggest the authors to keep separately the conclusions and the biosynthetic discussion.

Author Response

Note: Minor points the authors should modify:

1-Line 230, aglycones, please fix: replaced by the words “structures of the aglycones”

2-My main concern is about organization of the paper regarding the biogenesis discussion part. When I saw the title, it includes clearly 3 main points: structure, activity and biogenesis. However, the biogenesis part in the paper is included in the conclusions. I would suggest the authors to move it to a different previous section where the authors focus only in the biosynthesis. I would suggest the authors to keep separately the conclusions and the biosynthetic discussion.

Reply: According to the comment, additional section “2.3. Biogenesis of chilensosides A–G” was organized, where discussion of biogenesis moved from the Conclusions, the text was corrected (the changes are highlighted by yellow).

Reviewer 3 Report

The current MS can not be published in its current form. The below issues should be carefully addressed;

1- The title should be corrected to ``Chilensosides E, F, and G – New Tetrasulfated Triterpene .................................``

2- Remove, ``in continuation of our investigation``، from the abstract and rephrase.

3- Cucumber family name should be added in the abstract and keywords

3- English editing is needed there are many grammatical and typing mistakes also the proper verb tenses should be used.

4- less about the biogenesis should be added in the abstract.

5- Results of cytotoxic activity of the compound, type of assay and control results, and type of assay.

4- section 2 subheading should be modified to purification and structure elucidation of glycosides

5- Discussion of compounds needs improvement، For compound 1 aglycone should be discussed in detail regarding all spectral evidence including IR, 1D and 2D NMR COSY, HSQC, HMBC, then compounds 2 and 3 should be discussed in relation to compound 1.

6- Structures of chilensosides A and B should be added in figure 1.

7- In the general experimental section there are no verbs. The purpose of each mentioned tool should be added. 

8-the specimen number should be added.

9- A figure representing the 2DNMR correlations for both aglycone and sugar parts should be added in the discussion.

10_ Are the obtained results of biological activity agreed with the published data, highlight that in the discussion.

11- ``The procedures of separation, purification and isolation of the subfractions II (286 268 mg) and III.2 (177 mg) have already been discussed [15]`` this should be discussed in brief.

12- The IR spectrum should be measured and the data should be added and discussed in the MS.

13- A table for the NMR data if aglycone s should be included in the MS not in supporting materials. 

14- References for the procedure if the biological activity should be added.

15- References supported for the proposed biogenesis should be added.

Author Response

Note: 1- The title should be corrected to ``Chilensosides E, F, and G – New Tetrasulfated Triterpene .....”

Reply: fixed.

Note: 2- Remove, ``in continuation of our investigation`` from the abstract and rephrase.

Reply: Fixed, highlighted.

Note: 3- Cucumber family name should be added in the abstract and keywords.

Reply: Done, highlighted.

Note: 3- English editing is needed there are many grammatical and typing mistakes also the proper verb tenses should be used.

Reply: Extensive English edition was made.

Note: 4- less about the biogenesis should be added in the abstract.

Reply: The discussion of biogenesis in the abstract was shortened.

Note: 5- Results of cytotoxic activity of the compound, type of assay and control results, and type of assay.

Reply: Lines 213 – 215 were added: The earlier tested chitonoidoside L [3] was used as the positive control in all the tests (Table 4). The activity of the glycosides against SH-SY5Y, HeLa, and DLD-1 cells were examined by MTT assay, and against HL-60 and THP-1 cells – by MTS assay.

Note: 4- section 2 subheading should be modified to purification and structure elucidation of glycosides.

Reply: The subheading 2.2 was modified to “Structure elucidation”. We think the term “purification” shouldn’t be included to the title of 2.2 because it’s mainly devoted to structure elucidation and the isolation is discussed in general terms only. The purification and isolation are discussed in details in section 3.3.

Note: 5- Discussion of compounds needs improvement. For compound 1 aglycone should be discussed in detail regarding all spectral evidence including IR, 1D and 2D NMR COSY, HSQC, HMBC, then compounds 2 and 3 should be discussed in relation to compound 1.

Reply: The discussion of structure elucidation of the aglycone moiety of 1 was added and Table S1 was shifted from Supporting materials and renamed by Table 1 (Lines 85–99). The numbering of the following tables in manuscript and Supporting materials was changed (highlighted).

Note: 6- Structures of chilensosides A and B should be added in figure 1.

Reply: Chilensosides A and B are the previously published compounds (reference 15) and we think their adding to the Figure 1 together with new compounds discussed in the manuscript will mislead the readers.

Note: 7- In the general experimental section there are no verbs. The purpose of each mentioned tool should be added.

Reply: Fixed (lines 267–273).

Note: 8- The specimen number should be added.

Reply: The specimen number (36) is added. The voucher specimen number was added also as PIBOC-2019-MES-0135.

Note: 9- A figure representing the 2DNMR correlations for both aglycone and sugar parts should be added in the discussion.

Reply: Figure 2 is added, number of following figure is changed.

Note: 10- Are the obtained results of biological activity agreed with the published data, highlight that in the discussion.

Reply: Corresponding data are discussed (lines 238–241 and 253–254) and the references [19, 20] are added.

Note: 11-``The procedures of separation, purification and isolation of the subfractions II (286 268 mg) and III.2 (177 mg) have already been discussed [15]`` this should be discussed in brief.

Reply: Lines 307–312 containing the discussion of corresponding steps of purification are added.

Note: 12- The IR spectrum should be measured and the data should be added and discussed in the MS.

Reply: From the IR spectra the information concerning the presence of hydroxyls, carbonyl and sulfur can be obtained. We consider this method non-informative and not suitable for the structure elucidation due to more precise and modern methods were applied, such as NMR spectroscopy allowing to evidence the presence of these groups based on the chemical shift values and high-resolution mass spectrometry that gave exact mass of the compounds.

Note: 13- A table for the NMR data if aglycone s should be included in the MS not in supporting materials.

Reply: Table 1 with NMR data of the aglycone of chilensoside E was added.

Note: 14- References for the procedure if the biological activity should be added.

Reply: References [21–22] were added in the section “3.6 Hemolytic activity”, the methods MTT and MTS were performed in according to the protocol of manufacturer that doesn’t imply any references.

Note: 15- References supported for the proposed biogenesis should be added.

Reply: We referred to the paper [18] as well as the reference [21] is added.

Reviewer 4 Report

The manuscript entitled (Chilensosides E, F, and G – Tetrasulfated Triterpene Glycosides from the Sea Cucumber Paracaudina chilensis (Caudinidae, Molpadida): Structures, Activity and Biogenesis) by Silchenko et al. three new tetrasulfated triterpene glycosides, chilensosides E (1), F (2), and G (3) from the Far-Eastern sea cucumber Paracaudina chilensis. The structures were established on the basis of extensive analysis of 1D and 2D NMR spectra and confirmed by HR-ESI-MS data. Cytotoxicity and hemolytic activities of the isolated metabolites were evaluated. This manuscript could be accepted after the following corrections.

1-     The title should modify to (Chilensosides E–G: New Tetrasulfated Triterpene Glycosides from the Sea Cucumber Paracaudina chilensis).

2-     The abstract needs revision. It should be summarized and be concise, also some of your biological data should be included in the abstract.

3-     In keywords: add hemolytic.

4-     The MS is poorly written there are different typing and grammatical mistakes that should be corrected throughout the manuscript.

5-     In introduction authors should add previously metabolites isolated from the Sea Cucumber Paracaudina chilensis, along with their bioactivities.

6-     Authors said in discussion of compounds, but there is no any discussion for aglycone part which is similar in the three compounds, so please discuss the aglycone part in compound 1 extensively based on your NMR data.

7-     IR should be measured and included in the discussion at least for one of these compounds, especially compound number 3 as it was isolated in sufficient amount.

8-     What about the tested hemolytic activity results and its discussion, this should be included.

9-     The biosynthetic pathways chart of these metabolites and its related part should be moved to discussion part.

10- In extraction and isolation, please add Rt for each compound isolated by HPLC.

Author Response

Note: 1-      The title should modify to (Chilensosides E–G: New Tetrasulfated Triterpene Glycosides from the Sea Cucumber Paracaudina chilensis).

Reply: Fixed.

Note: 2-     The abstract needs revision. It should be summarized and be concise, also some of your biological data should be included in the abstract.

Reply: Abstract was shortened as result of revision, the broadening of discussion of biological data seems to be senseless because of the absence of cytotoxicity.

Note: 3-     In keywords: add hemolytic.

Reply: Done.

Note: 4-     The MS is poorly written there are different typing and grammatical mistakes that should be corrected throughout the manuscript.

Reply: Extensive English edition was made.

Note: 5-     In introduction authors should add previously metabolites isolated from the Sea Cucumber Paracaudina chilensis, along with their bioactivities.

Reply: Lines 45–49 discussing in brief the previously isolated metabolites and their activity are added to Introduction.

Note: 6-     Authors said in discussion of compounds, but there is no any discussion for aglycone part which is similar in the three compounds, so please discuss the aglycone part in compound 1 extensively based on your NMR data.

Reply: The discussion of structure elucidation of the aglycone moiety of 1 was added and Table S1 was replaced from Supporting materials and renamed by Table 1 (Lines 85–99). The numbers of the following tables in manuscript and Supporting materials were changed (highlighted).

Note: 7-     IR should be measured and included in the discussion at least for one of these compounds, especially compound number 3 as it was isolated in sufficient amount.

Reply: From the IR spectra the information concerning the presence of hydroxyls, carbonyl and sulfur can be obtained. We consider this method non-informative and not suitable for the structure elucidation due to more precise and modern methods were applied, such as NMR spectroscopy enabling to evidence the presence of these groups based on the chemical shift values and high-resolution mass spectrometry that gave exact mass of the compounds.

Note: 8-     What about the tested hemolytic activity results and its discussion, this should be included.

Reply: Some phrases: “The erythrocytes are traditional and convenient model for the investigation of membranolytic action of the glycosides. All the compounds 13 showed moderate hemolytic activity, allowing to suppose the cytotoxic doses of the investigated compounds will be rather high. Actually, chilensosides E–G (13) were not cytotoxic against cancer cell lines even at maximal tested concentration (100 µM).” were added to the text.

Note: 9-     The biosynthetic pathways chart of these metabolites and its related part should be moved to discussion part.

Reply: According to the comment, additional section “2.3. Biogenesis of chilensosides A–G” was organized, where discussion of biogenesis moved from the Conclusions, the text was corrected (the changes are highlighted by yellow).

Note: 10- In extraction and isolation, please add Rt for each compound isolated by HPLC.

Reply: HPLC retention times were added for chilensosides E–G.

Round 2

Reviewer 3 Report

No comment

Reviewer 4 Report

No comments